# Posiphen Reduces the Levels of Huntingtin Protein through Translation Suppression

**DOI:** 10.3390/pharmaceutics13122109

**Published:** 2021-12-07

**Authors:** Xu-Qiao Chen, Carlos A. Barrero, Rodrigo Vasquez-Del Carpio, E. Premkumar Reddy, Chiara Fecchio, Salim Merali, Alessia Deglincerti, Cheng Fang, Jack Rogers, Maria L. Maccecchini

**Affiliations:** 1Department of Neurosciences, University of California San Diego, La Jolla, CA 92093, USA; 2Department of Pharmaceutical Science, Moulder Center for Drug Discovery, School of Pharmacy, Temple University, Philadelphia, PA 19140, USA; chiarafecchio86@gmail.com (C.F.); smerali@temple.edu (S.M.); 3Department of Oncological Sciences, Icahn School of Medicine at Mount Sinai, New York, NY 10029, USA; rvdelcarpio@gmail.com (R.V.-D.C.); reddy.ep@gmail.com (E.P.R.); 4Sandoz, Novartis Division, Princeton, NJ 08540, USA; 5Laboratory of Stem Cell Biology and Molecular Embryology, The Rockefeller University, New York, NY 10065, USA; adeglinc@gmail.com; 6Annovis Bio, Berwyn, PA 19312, USA; fang@annovisbio.com (C.F.); maccecchini@annovisbio.com (M.L.M.); 7Neurochemistry Laboratory, Psychiatry-Neuroscience, Massachusetts General Hospital, Charlestown, MA 02129, USA; jrogers@partners.org

**Keywords:** Posiphen, stable-isotope labeling by amino acids in cell culture, proteomics, huntingtin, iron response elements, iron regulatory protein 1

## Abstract

Posiphen tartrate (Posiphen) is an orally available small molecule that targets a conserved regulatory element in the mRNAs of amyloid precursor protein (APP) and α-synuclein (αSYN) and inhibits their translation. APP and αSYN can cause neurodegeneration when their aggregates induce neurotoxicity. Therefore, Posiphen is a promising drug candidate for neurodegenerative diseases, including Alzheimer’s disease and Parkinson’s disease. Posiphen’s safety has been demonstrated in three independent phase I clinical trials. Moreover, in a proof of concept study, Posiphen lowered neurotoxic proteins and inflammatory markers in cerebrospinal fluid of mild cognitive impaired patients. Herein we investigated whether Posiphen reduced the expression of other proteins, as assessed by stable isotope labeling with amino acids in cell culture (SILAC) followed by mass spectrometry (MS)-based proteomics. Neuroblastoma SH-SY5Y cells, an in vitro model of neuronal function, were used for the SILAC protein profiling response. Proteins whose expression was altered by Posiphen treatment were characterized for biological functions, pathways and networks analysis. The most significantly affected pathway was the Huntington’s disease signaling pathway, which, along with huntingtin (HTT) protein, was down-regulated by Posiphen in the SH-SY5Y cells. The downregulation of HTT protein by Posiphen was confirmed by quantitative Western blotting and immunofluorescence. Unchanged mRNA levels of HTT and a comparable decay rate of HTT proteins after Posiphen treatment supported the coclusion that Posiphen reduced HTT via downregulation of the translation of *HTT* mRNA. Meanwhile, the downregulation of APP and αSYN proteins by Posiphen was also confirmed. The mRNAs encoding HTT, APP and αSYN contain an atypical iron response element (IRE) in their 5′-untranslated regions (5′-UTRs) that bind iron regulatory protein 1 (IRP1), and Posiphen specifically bound this complex. Conversely, Posiphen did not bind the IRP1/IRE complex of mRNAs with canonical IREs, and the translation of these mRNAs was not affected by Posiphen. Taken together, Posiphen shows high affinity binding to the IRE/IRP1 complex of mRNAs with an atypical IRE stem loop, inducing their translation suppression, including the mRNAs of neurotoxic proteins APP, αSYN and HTT.

## 1. Introduction

Posiphen ((+)-phenserine) is the chirally pure positive (+) enantiomer of (−)-phenserine (Phenserine) [1]. It is an orally bioavailable small molecule derived via a biochemical synthetic pathway. Posiphen is stable at room temperature. Thus far in the development program, Posiphen as the tartrate salt (Figure 1A) has been used in preclinical and clinical studies as a drug candidate for the treatment of neurological diseases associated with neurotoxic aggregating proteins. Unlike (−)-phenserine, it is not an acetyl-cholinesterase inhibitor [2]. Rather, it is an inhibitor of the translation of specific mRNAs and reduces the levels of at least two proteins, amyloid precursor protein (APP) and α-synuclein (αSYN) [3,4]. These proteins contribute to the formation of neurotoxic aggregates, called senile plaques and Lewy bodies, involved in the pathogenesis of Alzheimer’s disease (AD) and Parkinson’s disease (PD), respectively. Currently, Posiphen product is prepared as an oral dosage form, in hard gelatin capsules containing Posiphen as the tartrate salt, without excipients or fillers.

Posiphen’s effects have been described in multiple biological systems including in vitro cell models, animal models, and early phase human studies. Posiphen treatment reduced the levels of APP, along with its product Aβ42, and αSYN in human neuroblastoma cell cultures and rodent primary neurons [2,3,5,6]. Posiphen reduced APP and/or Aβ42 levels in the cerebral cortex of wild-type mice [2] and transgenic mice over-expressing the human *APP* gene with the Swedish mutation K670N/M671L (APP_SWE_), a model of early-onset AD [7]. Posiphen also lowered and normalized the levels of APP, its C-terminal fragment products and Aβ42 in the brain of the Ts65Dn mouse model of Down syndrome (DS) [8,9]. In the APP/PS1 mouse model of AD, Posiphen inhibited the translation of *APP* mRNA and normalized impairments in spatial working memory, contextual fear learning, and synaptic function [10]. In two mouse models of early PD, Posiphen normalized impaired colonic motility [11]. Posiphen rescued the aberrant early endosome phenotypes in the Ts65Dn mouse model of DS. It also restored axonal transport of neurotrophic signals, reduced phosphorylated tau (*p*-tau) and recovered the levels of choline acetyltransferase in this mouse model of DS [8]. Posiphen treatment proved well tolerated in three independent phase I clinical trials [12]. Most importantly, Posiphen reduced the levels of the soluble APP (sAPP) fragments, Aβ42, tau, and *p*-tau in the cerebrospinal fluid (CSF) of mildly cognitively impaired (MCI) patients [12].

Posiphen’s working mechanism involves a homologous iron response element (IRE) in the 5′-untranslated region (5′-UTR) of *APP* and *SNCA* (gene for αSYN) mRNAs [5,13,14]. IREs are conserved RNA stem-loop structures located in the 5′-or 3′-UTRs of mRNAs and involved in iron metabolism [15]. Iron regulatory proteins (IRPs) 1 and 2 post-transcriptionally control mammalian iron homeostasis by binding IREs. In the case of *APP* and *SNCA* mRNAs, IRP1 binding to the IRE is thought to prevent association with the ribosome, thus down-regulating translation [4,13,16]. In the presence of iron, IRP1 dissociates from mRNA and translation of the mRNA is upregulated. It was speculated that by increasing the affinity of IRP1 to IRE in the presence of iron, such as for the mRNAs of *APP* and *SNCA* [3,4,5,13,16,17,18], Posiphen inhibits their translation. Consistently, the potency of Posiphen for inhibiting the expression of *SNCA* was enhanced in the presence of iron [5]. Iron accumulation is thought to play a central role in the pathogenesis of neurodegenerative diseases, including PD and AD [19,20,21,22]. Iron increases the release of the mRNAs for neurotoxic proteins from IRP1, leading to their overexpression and neurodegeneration.

The mRNAs encoding other neurotoxic aggregating proteins, like prion mRNA, have 5′-UTRs with IREs homologous to the atypical IREs observed in *APP* and *SNCA* mRNAs [23], suggesting that additional mRNA (s) may also be targets of Posiphen. In the current study, we used the non-biased method of stable isotope labeling with amino acids in cell culture (SILAC) coupled with mass spectrometry (MS)-based proteomics to explore additional effects of Posiphen on protein expression. This method strongly identified downregulation of huntingtin (HTT) protein and the huntingtin signaling pathway. We confirmed the downregulation of HTT by quantitative Western blotting and immunofluorescence. Additionally, sequence alignment analysis identified that an atypical IRE exists in the 5′-UTR of HTT mRNA which bound IRP1. IRP1 bound both atypical and canonical IREs, but only the atypical IRE/IRP1 complex bound Posiphen specifically with high affinity. This work expands the working target spectrum of Posiphen and raises the possibility of using Posiphen to reduce mutant HTT (mHTT) in Huntington disease (HD).

## 2. Materials and Methods

### 2.1. Chemicals and Reagents

Stable isotope labeled (heavy) lysine and arginine (([13]C_6_, [15]N_2_)-l-lysine and ([13]C_6_)-l-arginine) were obtained from Cambridge Isotope (Andover, MA, USA) and normal (light) amino acids (l-lysine and l-arginine) were obtained from Sigma-Aldrich (St. Louis, MO, USA). All components of cell culture media were obtained from Thermo Fisher Scientific (Waltham, MA, USA), and protease inhibitor cocktail was obtained from Sigma-Aldrich (St. Louis, MO, USA). SILAC DMEM Medium (88364) and the dialyzed fetal bovine serum (FBS) (26400044) were purchased from Thermo Fisher Scientific. Trypsin (V5280) was purchased from Promega (Madison, WI, USA). All the chemicals for SILAC-MS were HPLC-grade unless specifically mentioned. DMEM: F12 50/50 mix medium was received from Corning (10-090-CV; Corning, NY, USA). FBS for regular culture of SH-SY5Y cell was from Omega Scientific (FB-02; Tarzana, CA, USA).

### 2.2. Cell Culture and Posiphen Treatment

The human neuroblastoma cell line SH-SY5Y (ATCC^®^ CRL 2266™) was obtained from American Type Culture Collection (Manassas, VA, USA). For SILAC-MS/MS analysis, cells were grown with SILAC media supplemented with 10% dialyzed fetal bovine serum, ([13]C_6_)-l-arginine, and ([13]C_6_, [15]N_2_)-l-lysine or normal L-arginine and l-lysine, to produce “heavy” or “light” SILAC medium, respectively. Cells were grown in parallel in either light or heavy media for 5 days, with media replacement every 24 h. Upon 90% confluent, heavy media fed cells were treated with Posiphen at 10 μM for 48 h while light media fed cells were treated with vehicle for 48 h.

For biochemistry analysis, SH-SY5Y cells were cultured in DMEM: F12 50/50 mix medium containing 10% FBS supplement plus penicillin and streptomycin at 37 °C in humidified atmosphere with 5% CO_2_. Once cells were 90% confluent, they were treated with vehicle or indicated concentrations of Posiphen for 48 h. Cells were washed briefly with cold phosphate-buffered saline (PBS) and lysed in PBS containing 1% NP-40, 0.1% SDS, 0.1% deoxylcholate, 0.2 mM sodium orthovanadate and protease inhibitor cocktail. All lysates were centrifuged (12,000 rpm for 15 min at 4 °C) to produce supernatants, which were then boiled in SDS-PAGE sample buffer.

The human embryonic fibroblasts came from the Coriell Institute (Camden, NJ, USA) and were cultured according to the manufacturer’s instructions. The RUES2 and RUES2-Q150 human embryonic stem cell lines were generated in the Brivanlou Lab [24]. These hESC lines were maintained under pluripotency conditions in mouse embryonic fibroblast-conditioned medium in the presence of 20 ng/mL basic fibroblast growth factor. Medium was changed every day and cells were passaged once every three days to avoid overcrowding. For Posiphen treatment, cells were passaged and treated with Posiphen at various concentrations or vehicle for 48 h without changing medium until harvesting and processing as shown for SH-SY5Y cells as above.

### 2.3. Sample Preparation, 1-D SDS-PAGE Separation and In-Gel Trypsin Digestion for SILAC Analysis

Proteins were processed for gel electrophoresis-liquid chromatography-mass spectroscopy (GeLC-MS)/MS proteomics analysis as previously described [25,26]. Total SH-SY5Y cell proteins were extracted from cells using RIPA buffer (25 mM Tris-HCl pH 7.6, 150 mM NaCl, 1% NP-40, 1% sodium deoxycholate, 0.1% SDS). Protein quantification was performed using Bradford method (Bio-Rad Protein Assay, Bio-Rad Laboratories, Hercules, CA, USA). 60 µg of total proteins (30 μg “heavy” and 30 μg “light”) were mixed and diluted with Laemmli sample buffer (Bio-Rad Laboratories, Hercules, CA, USA) containing 5% β-mercaptoethanol.

The protein mixture was heated for 5 min at 90 °C and loaded onto a 10% polyacrylamide gel. 1-D SDS-PAGE separation was performed using a mini-Protean II system (Bio-Rad Laboratories, Hercules, CA, USA) at 200 V for 45 min. Bands were visualized with SYPRO^®^ Ruby protein gel stain (S12001, Thermo Fisher Scientific, Waltham, MA, USA) for image acquisition, followed by SimplyBlue SafeStain (LC6065, Thermo Fisher Scientific, Waltham, MA, USA), and lanes were sliced into 10 sections, which were diced into ~ 1 × 1 mm^2^. After de-staining with 50% (*v/v*) acetonitrile (ACN) in 25 mM ammonium bicarbonate buffer (bicarbonate buffer), proteins within gel pieces were reduced with 10 mM dithiothreitol (DTT) in bicarbonate buffer and alkylated by incubation with 50 mM iodoacetamide in bicarbonate buffer. After gel dehydration with 100% ACN, the gel pieces were covered with approximately 40 µL of 12.5 µg/mL trypsin in bicarbonate buffer. In gel digestion was done at 37 °C for 12 h, trypsin was inactivated with formic acid at 2% final volume, and peptides were extracted and cleaned-up using the stage-tip clean-up method [27].

### 2.4. GeLC-MS/MS and Data Analysis

The desalted SILAC-labeled peptides were dried in a vacuum centrifuge and re-solubilized in 10 μL of 0.1% (*v*/*v*) formic acid. The tryptic peptide samples were loaded onto an Acclaim PepMap 100 pre-column (2 cm × 75 μm, Thermo Fisher Scientific, Waltham, MA, USA) and separated using an Acclaim PepMap RSLC C18 column (15 cm × 75 μm, Thermo Fisher Scientific, Waltham, MA, USA) at 300 nL/min delivered by an Easy nLC system (Thermo Fisher Scientific, Waltham, MA, USA). A mobile-phase gradient was run using mobile phase A (water/0.1% formic acid) and B (80% ACN/0.1% formic acid), as follows: from 0 to 30 min, isocratic hold at 20% B; from 30−45 min, gradient of 20−40% B; from 45−55 min, gradient of 40−60% B; and, from 55−65 min, gradient of 60−95% B.

Nano-electrospray ionization (nano-ESI) tandem MS was performed using a Q Exactive Orbitrap mass spectrometer (Thermo Fisher Scientific, Waltham, MA, USA). ESI was delivered using a stainless-steel emitter (ID 30 μm, length 40 mm, Thermo Fisher Scientific, Waltham, MA, USA) at a spray voltage of −2000 V. Using automatic switching between MS and MS/MS modes, MS/MS fragmentation was performed on the ten most abundant ions on each spectrum using collision-induced dissociation with dynamic exclusion (excluded for 10.0 s after one spectrum). The complete system was fully controlled by Xcalibur software (Thermo Fisher Scientific, Waltham, MA, USA).

Mass spectra processing was performed using the Proteome Discoverer (version 1.4, Thermo Fisher Scientific, Waltham, MA, USA). For SILAC analysis, the SILAC-search protocol of Proteome Discoverer was used. The generated de-isotoped peak list was submitted to an in-house Mascot server 2.2.07 for searching against the Swiss-Prot database (release 2015_06, version 56.6, 536029 sequences). Mascot search parameters were set as follows: species: *H**omo*
*sapiens* (20233 sequences); enzyme: trypsin with maximal 2 missed cleavage; fixed modification: cysteine carboxymethylation; variable modification: methionine oxidation; 0.45 Da mass tolerance for precursor peptide ions; and 0.6 Da for MS/MS fragment ions. All peptide matches were filtered using an ion score cutoff of 30. SILAC ratios for heavy/light peptide pairs were calculated by the Proteome Discoverer software, using the ten most abundant proteins as a normalizer.

### 2.5. Ingenuity Pathway Analysis and Interaction Network Analysis

Quantified proteins with heavy/light ratios were converted to fold change using ingenuity pathway analysis (IPA) computational software (QIAGEN IPA, Content version: 24390178 (Release Date: 17 June 2015), QIAGEN, Hilden, Germany, http://www.ingenuity.com, accessed on 8 July 2015). Differentially expressed proteins with fold change ≥1.25 and ≤−1.25 were analyzed for clusters of biological functions, pathways, and upstream regulators analysis using IPA and applying Fischer’s exact test and Z-score. The effects of Posiphen treatment on the interaction networks of HTT, APP and αSYN were analyzed using the STRING database (https://string-db.org/ accessed on: 2 February 2021) for proteins interactions. The total of differentially expressed 1053 proteins (≥1.25-fold and ≤−1.25-fold changes) were queried for network visualization using Cytoscape v3.8.2 (Cytoscape, Boston, MA, USA) with a confidence score cutoff of 0.70 and maximum additional interactions of 5. The first interacting proteins for HTT, APP and αSYN were identified, and a second analysis was performed using only the interacting proteins identified. The specific network of each protein was performed using one maximum additional interaction and the target proteins were searched as the central nodes of the network.

### 2.6. Protein Degradation Assay

SH-SY5Y cells treated with 10 μM Posiphen or vehicle for 48 h along with 10 μg/mL cycloheximide (CHX, C4859, MilliporeSigma, St. Louis, MO, USA) for the durations indicated followed by cell lysis and Western blotting to analyze the degradation rate of HTT under either condition.

### 2.7. Western Blotting Analysis

Protein samples (20–40 µg) were separated on 10% polyacrylamide gels and transferred to nitrocellulose membranes. The membranes were blocked with 5% blotting-grade milk in TBST (phosphate-buffer saline (PBS, pH 7.4) containing 0.05% Tween-20) and then probed overnight at 4 °C with mouse anti-huntingtin (1:1000; MAB5492, MilliporeSigma, St. Louis, MO, USA), rabbit anti-huntingtin (1:1000; D7F7 XP, Cell Signaling Technology, Danvers, MA, USA), rabbit anti-APP (1:5000; A8717, MilliporeSigma, St. Louis, MO, USA), mouse anti-α-synuclein (1:1000; 610786, BD Biosciences, San Jose, CA, USA), mouse anti-β-actin (1:10,000; 60008-1-Ig, Proteintech, Rosemont, IL, USA), or mouse anti-vinculin (1:2000; 05–386, MilliporeSigma, St. Louis, MO, USA) antibody. Membranes were then washed with TBST and incubated with HRP-conjugated secondary antibodies. All blots were developed using the BioRad Clarity Western ECL substrate and captured using ChemiDoc XRS + (Bio-Rad Laboratories, Hercules, CA, USA); only blots within signals in the linear range were quantitated using the ImageLab 3.0.1 software (Bio-Rad Laboratories, Hercules, CA, USA).

### 2.8. Immunofluorescence

SH-SY5Y cells grown under the conditions described above on poly-L-lysine-coated bottom glass 384 well plates (Greiner Bio-One, Kremsmünster, Austria; 781936), and treated with vehicle or Posiphen (10 µM) for 48 h. After the incubation time, cells were fixed in 4% paraformaldehyde in PBS for 20 min at room temperature, and ice-cold methanol for 15 min. Cells were permeabilized with 0.5% Triton X-100 and 0.5% Tween-20 in PBS, followed by incubation with an anti-HTT antibody (1:200; 5656, Cell Signaling Technology, Danvers, MA, USA) for 1 h at 37 °C and overnight at 4 °C. Cells were then washed in 0.5% Triton X-100 in PBS (3 times; 5 min/each) and incubated with goat anti-rabbit Alexa Fluor 647 secondary antibody (1:800; Thermo Fisher Scientific, Waltham, MA, USA) for 1 h at 37 °C. Nuclei were stained with DAPI (1 mg/mL; Thermo Fisher Scientific, Waltham, MA, USA) for 5 min. HTT signal was acquired in the deep red channel using the 40 ×water objective in an Operetta CLS (PerkinElmer, Waltham, MA, USA). Cells were identified using DAPI nuclear staining from 81 fields and 5 planes (1 µm distance from −2 to 2) stacks per well. Single-cell intensity of HTT staining was quantified using Harmony 4.8 software, and data was analyzed by Student’s *t*-test using Prism GraphPad 5 (Graphpad Software, San Diego, CA, USA).

### 2.9. mRNA Measurement

Total RNA was extracted from SH-SY5Y cells treated with Posiphen of different concentrations for 48 h using Quick RNA Miniprep Kit (Zymo Research, Irvine, CA, USA) and equal amounts of total RNA were used for cDNA generation with iScript^TM^ cDNA Synthesis Kit (Bio-Rad Laboratories, Hercules, CA, USA) following the manufacturer’s instructions [8]. The primer sequences used for human HTT were: fwd: 5′-GCTCTTAGGCTTACTCGTTCCT; rev: 5′-TTCCTTGTCACTCCGAAGCTG, from PrimerBank [28]. Polymerase chain reaction was for 40 cycles. Endogenous GAPDH mRNA was used as the internal control with the sequences: fwd: 5′-GCCACATCGCTCAGACACC; rev: 5′-AATCCGTTGACTCCGACCTTC. Values within the log-linear phase of the amplification curve were defined for each probe/primers set and analyzed using the ΔΔCt method (Applied Biosystems 7300 Real-Time PCR System, Applied Biosystems, Waltham, MA, USA).

### 2.10. RNA Sequence Alignment and RNA Secondary Structure Prediction

RNA sequences from the Huntingtin (*HTT*) gene were located using the NCBI Gene search and the Ensembl database. Since the 5′-UTR was of primary interest, the coding region was disregarded. The 5′-UTR region of *HTT* mRNA was aligned by the ClustalX2 graphical program to identify conservation (CAGUGN motif) with the canonical IREs of L-ferritin, H-ferritin, and ferroportin and the atypical IREs of *SNCA* and *APP* mRNAs. Then the RNA secondary structures were determined. Specifically, the CAGUGN motif and the RNA sequence extending from this motif in each IRE were used to generate the RNA stem-loops. Secondary structure folding of these RNA sequences was generated by the RNAFold webserver at the University of Vienna and was annotated using the RNAFold software package utilities. The RNAFold server provided the most probable secondary structure for *HTT* mRNA 5′-UTR sequence based on minimum free energy calculations [29].

### 2.11. IP-RT-PCR

As published before [16], SH-SY5Y cell lysates (500 μg) were incubated for 30 min at 4 °C in the presence of 30 μL of Dynabeads protein A (10001D, Thermo Fisher Scientific, Waltham, MA, USA). The lysates were then incubated for 3 h with agarose beads and 3 μg of either IRP1 (IRP11-A, Alpha Diagnostics International, San Antonio, TX, USA) or rabbit normal IgG (011-000-003, Jackson ImmunoResearch Laboratories, West Grove, PA, USA) at 4 °C. After washing five times, RNA was extracted from both supernatant and protein A beads preparations using Quick RNA Miniprep Kit as above. RT was performed using iScript^TM^ cDNA Synthesis Kit followed by PCR to detect the presence of mRNAs bound to IRP1 using gene-specific primer pairs of *HTT* primers (fwd, 5′-GCTCTTAGGCTTACTCGTTCCT; rev: 5′-TTCCTTGTCACTCCGAAGCTG and *APP* primers (fwd, 5′-GACAGACAGCACACCCTAAA; rev, 5′-CACACGGAGGTGTGTCATAA).

### 2.12. Microscale Thermophoresis

IRP1 was N-termini labeled with RED-NHS amino-reactive labeling kit (NanoTemper Technologies, München, Germany). Equilibrium binding experiments were performed by incubating N-termini-labeled protein (2 nM) with serial 2-fold dilutions of the IRE RNAs of H-Ferritin (GGGGUUUCCUGCUUCAACAGUGCUUGGACGGAACC) or APP (GGUGGCGGCGCGGGCAGAGCAAGGACGCGGCGGAU) at 125 µM for 10 min in incubation buffer (25 mM Tris pH 7.4, 150 mM NaCl and 0.1% Prionex). After incubation, samples were centrifuged at 13,000 rpm for 2 min before being loaded into standard capillaries provided by the manufacturer. Fluorescence values from the binding reactions were determined using the Monolith NT.115 (NanoTemper Technologies, München, Germany). Kd values were computed using the NanoTemper Technologies software. For determining Posiphen binding to the IRP1/mRNA complex, reactions containing IRP1 (2 nM)/H-ferritin (250 nM), or IRP1 (2 nM)/APP (600 nM)) were incubated for 10 min with 2-fold dilutions of Posiphen tartrate (50 µM). Samples were incubated and analyzed as described above. When experiments include multiple components, the order of addition matters for MST; IRP1 needs to be added to a solution that already contains IREs and Posiphen. RNAs were obtained from Dharmacon (Lafayette, CO, USA). RNAs were re-folded using a quick denaturation at 95 °C and then ice incubation for several min. Recombinant IRP1 was obtained from MyBiosource (San Diego, CA, USA).

### 2.13. Statistics

Data are presented as the mean ± SEM. Statistical analyses were performed using PRISM (GraphPad Software Inc., San Diego, CA, USA) with one-way ANOVA or Student’s *t*-test. The significance levels were: * *p* < 0.05, ** *p* < 0.01 and *** *p* < 0.001; n.s., non-significant.

## 3. Results

### 3.1. SILAC of Posiphen-Treated SH-SY5Y Cultured Neuroblastoma Cells

To explore altered proteins with decreased expression levels by Posiphen treatment, SH-SY5Y cells were subjected to SILAC coupled with LC-MS/MS proteomics analysis. Previous study has shown that Posiphen treatment at 10 μM for 48 h could induce considerable reduction of APP and αSYN in SH-SY5Y cells [3], thus we used the same condition in this study. Cells cultured in “heavy” medium were treated with Posiphen (10 μM), while cells cultured in “light” medium received the vehicle for 48 h. The two lysates were combined in equal amounts and then fractionated by SDS-PAGE. After in-gel digestion, proteins were identified and quantified by LC-MS/MS (Figure 1B). A total of 2463 proteins were detected with 136 up-regulated (≥1.25-fold change) and 917 down-regulated (≤−1.25-fold change) in SH-SY5Y cells treated with Posiphen (Appendix A Appendix A).

### 3.2. Functional Bioinformatic Characterization of Proteins Lowered by Posiphen Treatment

We then used Ingenuity Pathway Analysis (IPA) to identify canonical pathways, upstream regulators, and networks predicted from the differentially expressed proteins. The most significant canonical pathway (-log (*p*-value) > 1.3, i.e., *p* < 0.05) affected by Posiphen treatment was the HD signaling pathway (*p* = 0.0014) (Figure 1C; Appendix A Appendix A). Six proteins in this pathway were downregulated by Posiphen by more than 1.7 fold HTT, the adaptor related protein complex 2 alpha 2 (AP2A2), dynein cytoplasmic 1 intermediate chain (DYNC1I2), phospholipase C beta 3 (PLCB3), synaptosome associated protein 25 (SNAP25) and transcription elongation regulator 1 (TCERG1) (Figure 1D).

The upstream regulator analysis, with a focus on neurotoxic aggregating proteins, identified 15 down-regulated proteins with predicted upstream regulation by APP (Figure 2A,B). No readout was received for HTT and αSYN in the upstream regulator analysis. Additionally, network analysis identified several interacting proteins of HTT, APP and αSYN that were altered following Posiphen treatment. We identified eighteen interacting proteins for HTT, sixteen interacting proteins for APP and twenty-three interacting proteins for αSYN with a high confidence level (confidence score >70%). The expression fold change of these proteins by Posiphen treatment range from −4.256 to 2.76; most of them were down-regulated, represented as the blue nodes (Figure 2C; Appendix A Appendix A). Moreover, as expected among the down-regulated proteins in Posiphen-treated SH-SY5Y cells were APP and αSYN (Table 1). Furthermore, TAR DNA-binding protein 43 (TDP-43) was also downregulated by Posiphen (Table 1). To our knowledge, this is the first time that downregulation of HTT and TDP-43 by Posiphen treatment was observed.

### 3.3. Quantitative Confirmation of Posiphen-Downregulated Neurotoxic Aggregating Proteins

To confirm the downregulation of HTT, APP and αSYN after Posiphen treatment observed by SILAC as well as previous observations that Posiphen downregulates APP and αSYN proteins in several systems [2,3,5,6,8,11,12], the expression of these three proteins were quantified using Western blotting. Treatment of SH-SY5Y cells with Posiphen lowered HTT, APP and αSYN in a dose-dependent manner (Figure 3A–D). Statistically significant downregulation was observed by Posiphen at 10 μM for all the three proteins (Figure 3A–D). We also confirmed the downregulation of HTT by 10 μM of Posiphen in SH-SY5Y cells using immunofluorescence (Figure 3E). Next, we asked whether the *HTT* mRNA level was affected by Posiphen. To address this question, we quantitated the levels of *HTT* mRNA in the Posiphen-treated SH-SY5Y cells and no change in the levels of *HTT* mRNA was found (Figure 3F). To further rule out increased degradation of HTT protein in response to Posiphen, we examined its effects on the turnover of HTT in studies in which cycloheximide (CHX) was used to prevent protein synthesis. Posiphen had no effect on the rate of HTT degradation in SH-SY5Y cells and we saw a 50% reduction in HTT in both vehicle-and Posiphen-treated cells at about 24 h (Figure 3G), consistent with the published degradation rate of HTT [30].

To further confirm the downregulation of HTT by Posiphen, we treated wild type human fibroblasts and the WT-RUES2 and RUES2-Q150 pluripotent human stem cell lines with Posiphen of varying concentrations for 48 h. The RUES2-Q150 cell line was developed as a model of HD; it carries polyglutamine encoded by CAG repeat expansion in the *HTT* gene [24]. Posiphen treatment of all these three cell types decreased the levels of HTT at 10 μM (Appendix A Appendix A). These data support that Posiphen reduces HTT levels through downregulation of the translation of *HTT* mRNA.

### 3.4. Stem Loop Structures of Atypical and Canonical Iron-Responsive Elements

The mRNAs of L-and H-ferritin and ferroportin have a canonical IRE, while *APP* and *SNCA* mRNAs have an atypical IRE in their 5′-UTRs (Figure 4A,B). Since the SILAC proteomics showed that HTT level was lowered by Posiphen, we explored the presence of IRE in *HTT* mRNA 5′-UTR. We aligned the 5′-UTR of *HTT* mRNA with the other five 5′-UTRs containing an either canonical or atypical IRE. We identified a strong homology of the 5′-UTR of *HTT* mRNA with the “CAGAGC” sequence in the 5′-UTR of *APP* mRNA (Figure 4A). The detected IRE in the 5′-UTR of *HTT* mRNA was predicted to fold into an RNA stem-loop unlike the canonical IREs (Figure 4B) and more in line with an atypical IRE.

The atypical IREs of *APP* and *SNCA* mRNAs have been reported to bind IRP1 [16,31]. Next IP-RT-PCR was employed to test whether *HTT* mRNA interacts with IRP1 [16]. Applying an antibody specific to IRP1 or related control IgG to SH-SY5Y cell lysates, both the pull-down beads and the supernatant fractions were collected, followed by RNA extraction and reverse transcription-polymerase chain reaction (RT-PCR) with specific primers for *HTT* mRNA. We found that *HTT* mRNA bound to IRP1; as a positive control, *APP* mRNA also bound to IRP1 (Figure 4C). Altogether, *HTT* mRNA contains an atypical IRE in its 5′-UTR and can bind to IRP1.

### 3.5. Specificity of Posiphen Binding to Atypical IRE/IRP1 Complexes

Both canonical and atypical IREs have been reported to bind to IRP1 [16]. To understand the specificity of Posiphen’s interaction with these IRE/IRP1 complexes, equilibrium binding of IRP1 to IRE was assessed in the presence and absence of Posiphen using microscale thermophoresis (MST) binding experiments. We used H-ferritin IRE RNA and APP IRE RNA to represent canonical and atypical IREs, respectively. IRP1 was incubated with the H-ferritin IRE RNA or the APP IRE RNA sequences (Figure 5A). A fixed concentration of labeled IRP1 was used against increasing concentrations of RNA to obtain a dose-response/saturation curve. As expected from previous studies, the canonical IRE of H-ferritin and the atypical IRE of APP both bind IRP1 [16], with a somewhat higher affinity of the H-ferritin RNA/IRP1 complex (Kd = 88.5 nM) than the APP RNA/IRP1 complex (Kd = 225 nM).

The binding of Posiphen to IRP1/RNA complexes was also assessed at a fixed concentration of IRP1/RNA and increasing Posiphen concentrations, Posiphen failed to bind the RNA/IRP1 complex of H-ferritin, but bound the RNA/IRP1 complex of APP with high affinity (Kd = 3.2 nM, Figure 5B). Thus, Posiphen bound with specificity and high affinity to the APP RNA/IRP1 complex with an atypical IRE, but not to the H-ferritin RNA/IRP1 complex with a canonical IRE.

## 4. Discussion

To unbiasedly explore the proteins and pathways affected by Posiphen treatment, we applied SILAC proteomics to SH-SY5Y human neuroblastoma cells. We identified the HD signaling pathway, the upstream regulators of APP, and the HTT, APP and αSYN networks to be affected by Posiphen treatment. Consistent with these findings, Western blotting confirmed the downregulation of HTT, APP and αSYN in cells exposed to increasing concentrations of Posiphen. It is well established that Posiphen can reduce the expression of APP and αSYN [2,3,5,8,10,11,12], which was confirmed in this study. Interestingly, SILAC additionally showed for the first time that Posiphen reduced the expression of HTT and TDP-43 in SH-SY5Y cells. Altogether, these data support the previous data about Posiphen’s ability to reduce the expression of proteins linked to different neurodegenerative diseases.

We found that the three most affected pathways were Huntington’s disease signaling, DNA double strand break repair and phagosome maturation. Huntington’s disease signaling being the most affected top canonical pathway validated our finding that Posiphen inhibits neurotoxic proteins. Another major pathway, phagosome maturation, was also affected by Posiphen treatment. This led us to study the process in detail and let to the finding that phagosome maturation is tightly linked to intracellular transport. In neurons axonal transport was found to be impaired by pathogenic forms of APP and products, tau, aSYN, HTT, TDP43 and SOD1 [8,32,33,34,35,36]; all neurotoxic aggregating proteins, thus suggesting that these pathway changes might be downstream of neurotoxic protein changes.

A third pathway affected by Posiphen are DNA double-strand break and repair by homologous recombination, and granzyme A signaling. In fact, it has been established that DNA repair is impaired by high levels of Aβ and that in the brain deficient DNA repair may contribute to Alzheimer’s disease [37]. Although all of these findings are interesting and exciting, future studies are warranted to investigate the biological significance of these findings.

The surprisingly wide action of Posiphen on such functionally different and structurally unrelated proteins can be explained by the mechanism of action described for Posiphen. It is a translation inhibitor acting on the conserved 5′-UTR IREs of mRNAs encoding different neurotoxic proteins, including APP, αSYN, and HTT [2,3,4,5,8,11,12,13,16,17,18,23]. This element is the molecular target of Posiphen, which was speculated to increase the affinity of the IRE/IRP1 interaction, thereby attenuating the translation of mRNA harboring such IRE [3] (Figure 5C). This suggests that similar 5′-UTR IREs regulate the translation of these neurotoxic proteins.

Furthermore, the IRE of *APP* mRNA in the 5′-UTR was found recently to overlap with the active site for the microRNA miR-346, which was proven to upregulate APP translation. Under conditions of low cellular iron levels, miR-346 can inhibit IRP1 binding to the IRE and enhance APP translation [38]. On the other hand, increased iron levels could enhance APP mRNA translation by dissociating IRP1 with IRE [16] and Posiphen would oppose the impact of iron by recruiting IRP1 to IRE. Thus, iron load can affect not only the APP mRNA translation but also the effects of both miR-346 and Posiphen on APP mRNA translation. How Posiphen interacts with miR-346 regarding the APP mRNA translation is yet to be clarified.

In this study, we found that Posiphen treatment induced a reduction in the levels of HTT protein, which was confirmed in several cell lines. Posiphen had neither a significant effect on the levels of *HTT* mRNA nor the degradation rate of HTT protein, further supporting a translation-dependent mechanism for Posiphen. When the APP and H-ferritin IREs, representing atypical and canonical IREs respectively, were tested in MST binding assay. Although IRP1 can bind to both APP and H-ferritin IREs in MST binding assay, Posiphen only bound to the APP RNA/IRP1 complex with high affinity. These data are consistent with our findings that Posiphen reduced the levels of APP, αSYN and HTT but not L-and H-ferritin in the SILAC proteomics. Supporting that the working model for Posiphen to reduce the levels of neurotoxic proteins via potentiation of the binding of IRP1 to IRE present in the mRNAs of several neurotoxic proteins.

It is noted that the mutated HTT (mHTT) harboring polyglutamine encoded by CAG repeat expansion in *HTT* gene is responsible for HD [39]. Lowering such mHTT has been regarded as a possible route to reserve HD pathology, as shown by antisense oligonucleotide and the cytosolic chaperonin T-complex 1 ring complex subunit-mediated downregulation of mHTT [32,40]. The effects of Posiphen on HTT in our study raised the possibility to use Posiphen to reduce the levels of mHTT in HD subjects as mHTT harbors the same 5′-UTR as wild type HTT. Although we have shown the effect of Posiphen on mHTT in the RUES2-Q150 cell line harboring mHTT in this study, whether Posiphen is potent to reduce the levels of mHTT in vivo needs to be explored in the future. The in vivo environment may affect the potency of Posiphen on HTT as revealed in Ts65Dn mice in which Posiphen more potently downregulates APP compared with in its euploid control mice [8]. Consistently, like human *HTT* mRNA, mouse *HTT* mRNA also contains “CAGAGC” in the 5′-UTR. Additional preclinical studies using mHTT knockin/transgenic mouse models of HD will be needed to evaluate the effects of Posiphen on mHTT and accompanied HD pathologies. Furthermore, although we showed here that Posiphen reduced HTT in human-derived fibroblast and ESC cells, there is an urgent need to test the effects of Posiphen on HTT, especially mHTT in induced pluripotent stem cell-derived or induced human neurons. Based on literature and our current findings we speculate that Posiphen increases the binding affinity of IRP1 to IREs in the 5′-UTRs of mRNAs of several targeted proteins; however, the direct evidence supporting this is still lacking. Further studies are needed to examine the detailed mechanism of action for Posiphen.

## 5. Conclusions

Applying SILAC proteomics to SH-SY5Y human neuroblastoma cells, we confirmed the downregulation of APP and αSYN by Posiphen treatment. Remarkably, the most significantly affected pathway by Posiphen was the HD signaling pathway, which, along with HTT protein, was down-regulated by Posiphen. With quantitative Western blotting and immunofluorescence assays, we confirmed the downregulation of HTT by Posiphen. Further experiments unveiled that Posiphen reduced the levels of HTT in a translation-dependent manner. Bioinformatic analysis predicted that like *APP* and *SNCA* mRNAs, *HTT* mRNA contains an atypical IRE in its 5′-UTR that binds IRP1. The downregulation of HTT by Posiphen raised the possibility to use Posiphen to reduce the levels of mHTT in HD to combat mHTT-associated neuropathologies.

## Figures and Tables

**Figure 1 pharmaceutics-13-02109-f001:**
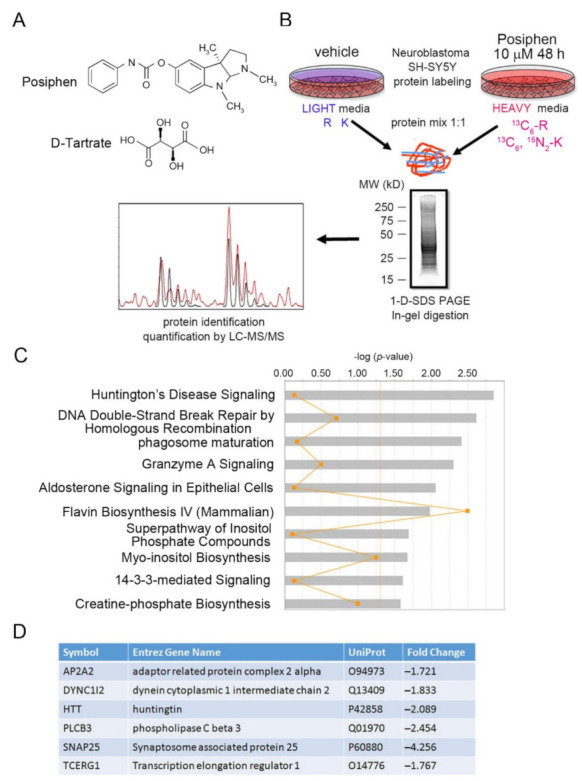
Main canonical pathways affected by Posiphen using the stable isotope labeling with amino acids in cell culture (SILAC) method for SH-SY5Y cells. (**A**) Chemical structure of Posiphen as the Tartrate salt. (**B**) Cell lysate from the two treatment conditions (with or without Posiphen 10 μM) were mixed in equal quantities and loaded on a gel. The obtained gel stained with SYPRO^®^ is reported. Gel was cut and subjected to tryptic digestion prior to analysis by mass-spectrometry in Nano LC-MS/MS Orbitrap Q-Exactive. Raw data were subjected to bioinformatics analysis to identify and subsequently filter proteins with significant changes in expression levels. Ingenuity Pathway Analysis (IPA) identified cluster pathways affected by the treatment. MW: molecular weight; SDS-PAGE: sodium dodecyl sulfate-polyacrylamide gel electrophoresis; LC-M/MS: liquid chromatography tandem mass spectrometry. (**C**) Differentially expressed proteins were clustered by IPA. The 10 most statistically significantly affected (*p* < 0.05) canonical pathways are shown. The significance threshold value (-log [*p*-value]), set at 1.3, is highlighted in orange. Orange pints are the ratio of the number of genes that meet the cut-off criteria/the number of genes that make up that pathway. (**D**) The most downregulated proteins (≤1.7) in the Huntington’s pathway are listed.

**Figure 2 pharmaceutics-13-02109-f002:**
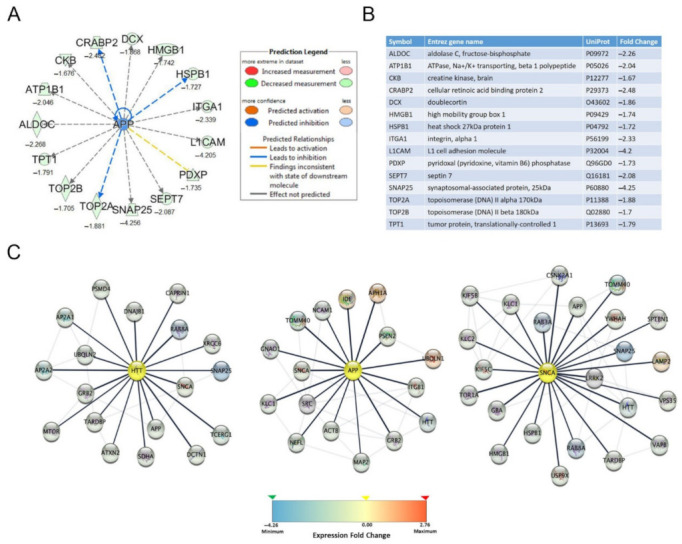
Functional bioinformatic characterization of proteins lowered by Posiphen treatment. (**A**,**B**) Up-stream regulator analysis by IPA identified APP related proteins. Several APP up-stream regulators were identified by IPA up-stream regulator analysis. Proteins with fold change ≥1.5 or ≤0.5 are listed in the table (**B**) and schematically represented (**A**). Legend key of IPA prediction is displayed. (**C**) Schematic representation of the interaction network of HTT, APP and αSYN. The central nodes of the networks are highlighted in yellow; the first interacting proteins are linked by the thick dark lines, and the additional interacting proteins within the network are represented by light gray lines. The scale of the protein expression levels is shown in a graded scale from down-regulated (**blue**) to up-regulated (**red**) proteins. The interacting proteins identified have a have a high confidence score (>70%).

**Figure 3 pharmaceutics-13-02109-f003:**
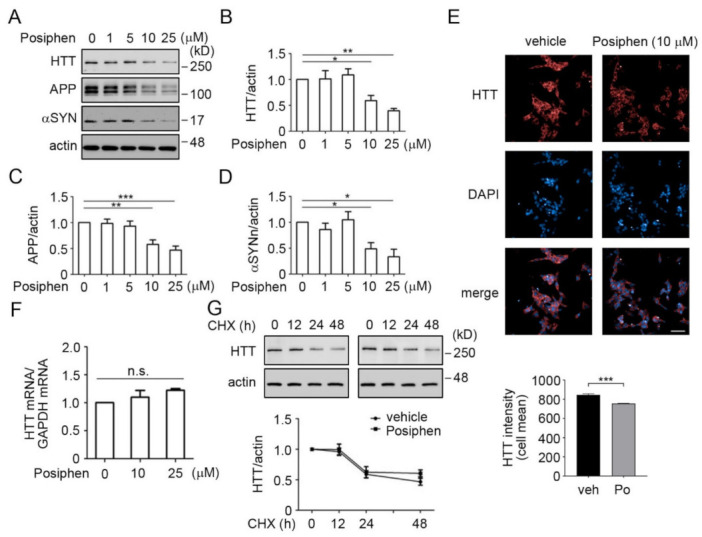
Posiphen reduced HTT levels in a dose-dependent manner in SH-SY5Y cells. (**A**) SH-SY5Y cells were treated with different concentrations of Posiphen, as indicated, for 48 h, followed by immunoblotting to analyze the levels of HTT, APP and αSYN. β-actin served as an internal loading control. The results of statistical analysis for each protein in (**B**–**D**) are shown (*n* = 3–4). (**E**) Representative confocal images of SH-SY5Y cells treated with Posiphen 10 µM or vehicle of four replicates and three independent assays, showing immunostaining against HTT (**red**). DAPI (**blue**) was used for nuclei detection (Scale bar: 50 µm). Single cell intensity of HTT staining was quantified. Histogram shows the mean (±SEM) of single cell intensity for Posiphen treatment vs. control (more than 10, 000 cells per group). Data was analyzed by Student’s *t* test. *** *p* < 0.001. (**F**) The levels of HTT mRNAs from SH-SY5Y cells treated with vehicle or Posiphen at 10 μM for 48 h were assessed via real-time PCR (*n* = 3). (**G**) The turnover rates of the HTT protein level in vehicle or Posiphen-treated SH-SY5Y cells were measured by 10 μg/mL CHX co-treatment for the durations indicated with the levels of HTT normalized to β-actin (*n* = 4). * *p* < 0.05, ** *p* < 0.01, *** *p* < 0.001, n.s., non-significant, one-way ANOVA followed by Newman–Keuls multiple comparison test for panels (**B**–**F**).

**Figure 4 pharmaceutics-13-02109-f004:**
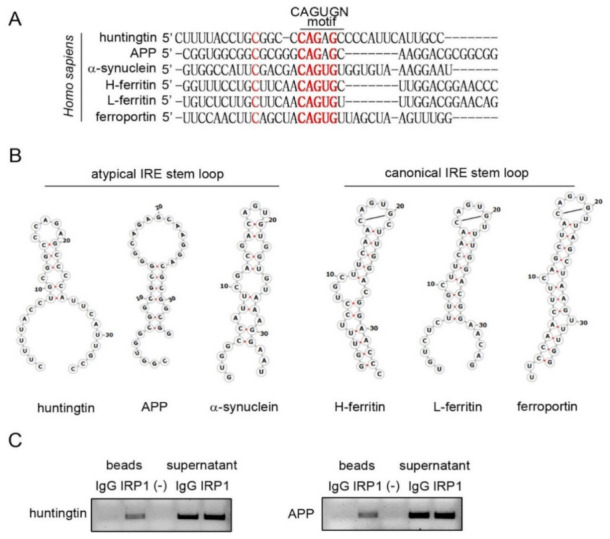
The 5′-UTR of *HTT* mRNA contains an atypical IRE Stem loop. (**A**) Alignment of the predicted IRE in the 5′-UTR of *HTT* mRNA against to the IREs of APP, αSYN, L-ferritin, H-ferritin and ferroportin mRNAs. Highlight in red are the conserved bases in the CAGUGN motif and an unpaired “bulge” cytosine. (**B**) The atypical 5′-UTR IRE stem loops of *APP*, *SNCA* (α-synuclein) and *HTT* mRNAs differ from canonical IREs found in L-and H-ferritin and ferroportin mRNAs. Both types of IREs, the canonical and atypical IREs, bind to IRP1. However, Posiphen only inhibits the translation of neurotoxic proteins whose mRNAs have an atypical IRE in the 5′-UTR. (**C**) Representative IP-RT-PCR experiment showing that *HTT* mRNA binds IRP1. SH-SY5Y cell lysates were incubated with either IPR1 specific antibody or control IgG and Dynabeads Protein A slurry. Following IP, RT-PCR was performed from RNAs extracted from both beads and supernatant fractions where specific HTT and APP primers were used to detect the presence/absence of each transcript (*n* = 3).

**Figure 5 pharmaceutics-13-02109-f005:**
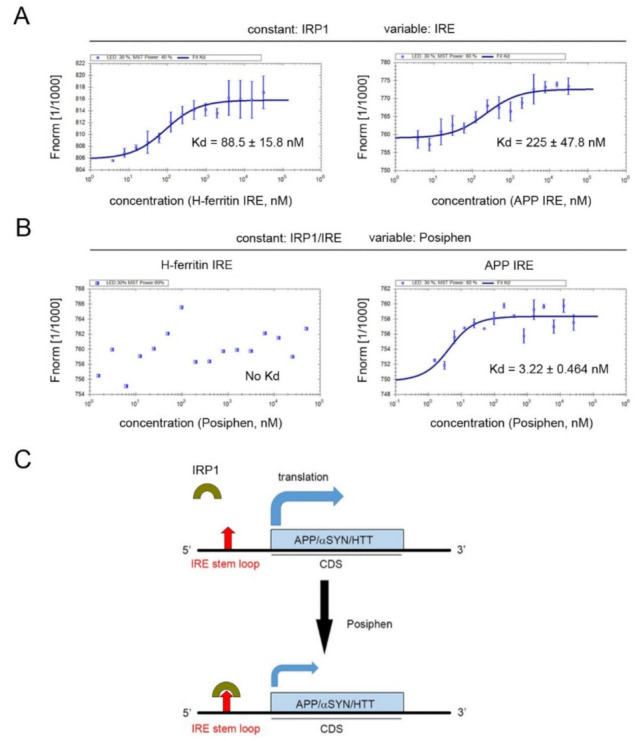
MST equilibrium binding assays of IRP1/RNA and Posiphen. (**A**) IRP1 was incubated with the H-ferritin canonical IRE RNA or the APP atypical IRE RNA sequences. A saturation curve was received using a fixed concentration of labeled IRP1 and increasing concentrations of RNA. IRP1 bound to H-ferritin IRE RNA (**left**) and to APP IRE RNA (**right**). (**B**) At a fixed concentration of IRP1 and IRE RNA of either H-ferritin or APP and increasing Posiphen concentrations, Posiphen only bound to the IRP1/APP IRE complex (**right panel**) but not to IRP1/H-ferritin IRE complex (**left**). (**C**) A working model for Posiphen to repress the translation of mRNA containing an atypical IRE in the 5′-UTR. IRP1 controls the expression of APP, αSYN and HTT whose mRNAs harbor an atypical IRE as well as L-and H-ferritin and ferroportin whose mRNAs contain a canonical IRE in an iron-dependent manner. Posiphen can bind to IRP1/atypical IRE complex, but not IRP1/canonical IRE complex and was predicted to potentiate the binding of IRP1 to IRE, thus further prohibiting 40S ribosome access to the 5′-UTRs of these mRNAs encoding APP, αSYN and HTT and reducing their translation. IRE: iron-responsive element; CDS: coding sequence.

**Table 1 pharmaceutics-13-02109-t001:** Changes in protein levels of neurotoxic proteins whose mRNAs contain an atypical IRE and control proteins whose mRNAs contain a canonical IRE (FTL, FHL1 and TFRC) in their 5′-UTRs following Posiphen treatment of SH-SY5Y cells.

Symbol	Entrez Gene Name	UniProt	Heavy/Light
HTT	Huntingtin	P42858	0.479
APP	Amyloid beta A4 protein	P05067	0.728
TARDBP	TAR DNA-binding protein 43	Q13148	0.761
SNCA	Alpha-synuclein	P37840	0.779
SOD1	Superoxide dismutase [Cu-Zn]	P00441	1.072
FTL	Ferritin light chain	P02792	0.978
FTH1	Ferritin heavy chain	P02794	0.975
TFRC	Transferrin receptor protein 1	P02786	0.874

## Data Availability

All data generated or analyzed during this study are included in this paper.

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
