# Peer review of "Posiphen Reduces the Levels of Huntingtin Protein through Translation Suppression"

_pharmaceutics, 2021, doi:10.3390/pharmaceutics13122109_

Round 1
Reviewer 1 Report
The manuscript proposed by Maccecchini and coworkers is about the investigation of the effects of Posiphen treatment on protein expression. It is a very interesting and well written work. The reduction of huntingtin (HTT) protein, as well as the downregulation of APP and alpha-SYN proteins by Posiphen has been demonstrated.
The manuscript can be accepted for publication on Pharmaceutics journal after minor revision:
I suggest the author to report in the SI the experimental conditions adopted for LC-MS analysis;
In paragraph 2.3 the unit for trypsin concentration has to be corrected;
Page 11: a space is missing: “wasemployed”.
Author Response
Comments and Suggestions for Authors
The manuscript proposed by Maccecchini and coworkers is about the investigation of the effects of Posiphen treatment on protein expression. It is a very interesting and well written work. The reduction of huntingtin (HTT) protein, as well as the downregulation of APP and alpha-SYN proteins by Posiphen has been demonstrated.
The manuscript can be accepted for publication on Pharmaceutics journal after minor revision:
I suggest the author to report in the SI the experimental conditions adopted for LC-MS analysis;
We thank the reviewer. The detailed protocol along with the experimental conditions has been described in 2.4 in Page 4.
In paragraph 2.3 the unit for trypsin concentration has to be corrected;
Thanks, we changed it.
Page 11: a space is missing: “wasemployed”.
Thanks, we’ve corrected it highlighted in Page 12.

Reviewer 2 Report
The manuscript pharmaceutics-1471553 "Posiphen Reduces the Levels of Huntingtin Protein through Translation Suppression" by Xu-Qiao Chen et al describes the Posiphen as a promising drug candidate for neurodegenerative diseases including Alzheimer’s disease and Parkinson’s disease. The manuscript is very well structured and presents complex studies about the Posiphen that it has already gone through phase I of clinical trials.
The authors can complete the presented information with some details about the Posiphen strucure, solubility, and stability.
The manuscript presents a great interest to the readers of Pharmaceutics, is well written and I suggest the publication in this journal.
Author Response
The manuscript pharmaceutics-1471553 "Posiphen Reduces the Levels of Huntingtin Protein through Translation Suppression" by Xu-Qiao Chen et al describes the Posiphen as a promising drug candidate for neurodegenerative diseases including Alzheimer’s disease and Parkinson’s disease. The manuscript is very well structured and presents complex studies about the Posiphen that it has already gone through phase I of clinical trials.
The authors can complete the presented information with some details about the Posiphen strucure, solubility, and stability.
We thank the reviewer and have enriched the related information in “introduction” section in Page 2.
The manuscript presents a great interest to the readers of Pharmaceutics, is well written and I suggest the publication in this journal.

Reviewer 3 Report
The manuscript describes the effect of posiphen treatment on SHSY-5Y cells. It shows very novelty results about the effectivity of the molecule posiphen in decreasing some Huntington's disease-associated proteins. Likewise, it offers data to relate the effect of posiphen to Huntington's disease signaling.
Why do you select SHSY-5Y cells to do the study?
Why do you use a different medium (heavy or light) on dependent of treatment of cells, and why, for proteomic analysis, do you mix the proteins.
I miss more detailed information about the posiphen molecule.
In figure 1, the first legend, a 's', is omitted (diseae instead of disease).
Author Response
Comments and Suggestions for Authors
The manuscript describes the effect of posiphen treatment on SHSY-5Y cells. It shows very novelty results about the effectivity of the molecule posiphen in decreasing some Huntington's disease-associated proteins. Likewise, it offers data to relate the effect of posiphen to Huntington's disease signaling.
Why do you select SHSY-5Y cells to do the study?
SH-SY5Y is a regular cell line derived from the SK-N-SH neuroblastoma cell line. It serves as a model for many neurodegenerative diseases as it can be converted to various types of functional neurons by the addition of specific compounds. The target proteins of Posiphen are all related to neurodegenerative diseases. Thus we usually choose SH-SY5Y cells to explore the effects of Posiphen. Actually, Posiphen has been tested in this type of cell for its effects on APP and alpha-synuclein before.
Why do you use a different medium (heavy or light) on dependent of treatment of cells, and why, for proteomic analysis, do you mix the proteins.
SILAC was reported to be the most accurate quantitative MS methods. The principle of SILAC is based on metabolically incorporating stable isotope labeled amino acids, like 13C or 15N-labeled arginine or lysine, into the proteome during protein metabolism, specifically in the process of cell culture. In SILAC, two groups of cells are grown in different culture media, with the “light” medium containing amino acid(s) with the natural isotope, and the “heavy” medium containing stable isotope labeled amino acid(s). After a sufficient number of cell divisions, theoretically all the proteins inside the cells cultured in heavy medium contain amino acids in the heavy state. After complete labeling (at least >95% labeling efficiency), the cell populations are experimentally manipulated and then equal amounts of labeled and unlabeled cells or protein extracts are mixed together as we described in the manuscript. The samples are then digested into peptides. Finally, the digested peptides are analyzed with LC-MS/MS. The quantification of SILAC is based on testing the ratio of introduced isotope-labeled peptides to unlabeled peptides. Thus, the signal intensities from light and heavy samples allow for quantitative comparison of their relative abundances in the mixture (Chen, 2015).
I miss more detailed information about the posiphen molecule.
We thank the reviewer and have enriched the related information in “Introduction” section in Page 2.
In figure 1, the first legend, a 's', is omitted (diseae instead of disease)
We thank the reviewer and corrected it in Fig. 1.

Reviewer 4 Report
the paper presented by Chen and coworkers is interesting and deserves publication in this journal. However, some modifications are required for improving the quality of the paper.
-style of the references is not in agreement with the journal guidelines. The template should be enclosed row numbers
-purity of chemicals used should be clearly stated
-number of independent experiments for each assays performed should be reported in the materials and methods
-check acronyms since some of them were repeated (e.g. Huntington’s disease (HD) huntingtin (HTT) etc that were defined at least two times). Please correct inserting the abbreviation at the first time where the terms appear.
-figure legends should be inserted after the figures
-limitations of the study can be introduced and how the study can be useful in the future should be discussed.
Author Response
Comments and Suggestions for Authors
the paper presented by Chen and coworkers is interesting and deserves publication in this journal. However, some modifications are required for improving the quality of the paper.
-style of the references is not in agreement with the journal guidelines. The template should be enclosed row numbers
We thank the reviewer and we noted that in "author instruction" references can be in any style provided they are consistent.
-purity of chemicals used should be clearly stated
Thanks. All the chemicals for SILAC-MS were HPLC-grade (Page 3). For other biochemistry and cell biology experiments, regular reagents were used as indicated.
-number of independent experiments for each assays performed should be reported in the materials and methods
We thank the reviewer and have indicated the number (n) in the figure legends.
-check acronyms since some of them were repeated (e.g. Huntington’s disease (HD) huntingtin (HTT) etc that were defined at least two times). Please correct inserting the abbreviation at the first time where the terms appear.
Thanks. We corrected it in 3.2.
-figure legends should be inserted after the figures
We thank the reviewer and have changed them.
-limitations of the study can be introduced and how the study can be useful in the future should be discussed.
We thank the reviewer and have enriched the discussion in Page 15 about the limitations and perspectives.

Reviewer 5 Report
Comunication by Xu-Qiao Chen et al. “Posiphen Reduces the Levels of Huntingtin Protein through Translation Suppression” aims to explore the drug posiphen. The manuscript is relevant, as it first revealed the effect of posiphen on huntingtin protein. There are practically no data on the effect of posiphen on HTT in the literature. This new information will be useful for determining the spectrum of action of posiphen
The authors note that, according to the literature, posiphen targets a conserved regulatory element in the mRNAs of amyloid precursor protein (APP) and α-synuclein (αSYN) and inhibits their translation. Most part of the article is devoted to confirming the already known data on the effect of posiphen on APP and αSYN. And only the third part describes the effect of posiphen on HTT. This is undoubtedly new and very interesting data. It seems to me that it would be correct to leave only data related to the effect of posiphen on HTT. This is "Comunication" after all, not "Research Article". This would greatly improve the readability of the manuscript. This is the main disadvantage of this work.
In addition, a number of minor comments can be made.
Abbreviations should be arranged alphabetically.
Sections 2.1; 2.2 Isotopes should be written in superscript.
Sample preparation is described only for SH-SY5Y cell, for fibroblasts, WT-RUES2 and RUES2-Q150 there is no sample preparation description.
Section 3.1. Mentioned table 1 in supplementary material. However, there are only Excel files, tables are present, but they do not have a name or a number.
Section 3.2. Table 3 is mentioned in the supplementary material, but there is no such table.
Supplementary file deserves a separate review. As presented, these are 3 Excel files and one Word. The first file is called "Supplementary Figure 1 Posiphen_SILAC_Raw data", while the Word file contains "Supplementary Figure" and contains Supplementary Figure 1, which has nothing to do with the first file. It is necessary not only to attach additional data, but also to explain how and why they were obtained, name the tables present in the files, make indications of these tables in the text (if they are significant).
Figure 1. The figure caption is after the figure. As it should be. It's not very clear why insert a table into a picture?
For all other figures, the caption is in front of the figure.
Figure 2. Why regulatory proteins are indicated for APP rather than HTT. Due to the fact that in Figure 2C all three studied proteins are indicated, the drawing is difficult to understand.
The Discussion states that "the effect of Posiphen on mHTT was verified in the RUES2-Q150 cell line in this study." However, the text of the article only contains a link to Figure 1 in the Supplementary files. There is no indication in the file what kind of data it is, what it refers to and how it was obtained.
Author Response
Comments and Suggestions for Authors
Comunication by Xu-Qiao Chen et al. “Posiphen Reduces the Levels of Huntingtin Protein through Translation Suppression” aims to explore the drug posiphen. The manuscript is relevant, as it first revealed the effect of posiphen on huntingtin protein. There are practically no data on the effect of posiphen on HTT in the literature. This new information will be useful for determining the spectrum of action of posiphen
The authors note that, according to the literature, posiphen targets a conserved regulatory element in the mRNAs of amyloid precursor protein (APP) and α-synuclein (αSYN) and inhibits their translation. Most part of the article is devoted to confirming the already known data on the effect of posiphen on APP and αSYN. And only the third part describes the effect of posiphen on HTT. This is undoubtedly new and very interesting data. It seems to me that it would be correct to leave only data related to the effect of posiphen on HTT. This is "Comunication" after all, not "Research Article". This would greatly improve the readability of the manuscript. This is the main disadvantage of this work.
We thank the reviewer. As all the three proteins APP, αSYN and HTT interact with each other as shown in Figure 2. In this study, we reported the effects of Posiphen on all these three proteins in parallel on one hand to confirm the previous findings with SILAC-MS, on the other hand to compare the potency of Posiphen on them and compare network regulation and IRE structure similarity. Thus we prefer to keep the existing data in this manuscript.
In addition, a number of minor comments can be made.
Abbreviations should be arranged alphabetically.
Thanks. We have changed it in Page 2.
Sections 2.1; 2.2 Isotopes should be written in superscript.
We thank the reviewer very much for pointing this out and we have corrected them also in Fig. 1.
Sample preparation is described only for SH-SY5Y cell, for fibroblasts, WT-RUES2 and RUES2-Q150 there is no sample preparation description.
Thanks. We added the related information in Page 3-4.
Section 3.1. Mentioned table 1 in supplementary material. However, there are only Excel files, tables are present, but they do not have a name or a number.
We thank the reviewer and changed its name to Supplementary Table 1.
Section 3.2. Table 3 is mentioned in the supplementary material, but there is no such table.
We thank the reviewer and changed its name to Supplementary Table 3.
Supplementary file deserves a separate review. As presented, these are 3 Excel files and one Word. The first file is called "Supplementary Figure 1 Posiphen_SILAC_Raw data", while the Word file contains "Supplementary Figure" and contains Supplementary Figure 1, which has nothing to do with the first file. It is necessary not only to attach additional data, but also to explain how and why they were obtained, name the tables present in the files, make indications of these tables in the text (if they are significant).
Sorry for the confusion. We renamed all the documents in Supplementary file. Now they are clear.
Figure 1. The figure caption is after the figure. As it should be. It's not very clear why insert a table into a picture?
For all other figures, the caption is in front of the figure.
Thanks. We changed them now. For the table in Figure 1, we want to display the change folds of these proteins after Posiphen treatment. If the reviewer requires to remove it, we will do.
Figure 2. Why regulatory proteins are indicated for APP rather than HTT. Due to the fact that in Figure 2C all three studied proteins are indicated, the drawing is difficult to understand.
We also did the upstream regulator analysis for the other two proteins, however no readout was received for HTT and αSYN in the upstream regulator analysis. We also pointed it in Page 9 highlighted in yellow.
The Discussion states that "the effect of Posiphen on mHTT was verified in the RUES2-Q150 cell line in this study." However, the text of the article only contains a link to Figure 1 in the Supplementary files. There is no indication in the file what kind of data it is, what it refers to and how it was obtained.
We highlighted the information about Supplementary Figure 1 in yellow in Page 11. We also renamed the Supplementary documents in the Supplementary files for clarification. There is a figure legend for Supplementary Figure 1 below the data.

Round 2
Reviewer 4 Report
Authors addressed my concerns in the revised version.
Reviewer 5 Report
I thank the authors who took into account all the comments made. I am confident that the manuscript has become more readable and attractive.
The article is ready for publication.